# Analysis of the Influence of Entrepreneurial Apprehension and Entrepreneurial Strategic Orientation on Breakthrough Innovation

**Qiliang Wang [1] , Qingquan Jiang [2],* and Hongxia Yu [3],***

1    College of Business, Quzhou University, Quzhou 324000, China; 40080@qzc.edu.cn
2    School of Economics & Management, Xiamen University of Technology, Xiamen 361024, China
3    School of Economics & Management, Zhejiang Normal University, Jinhua 321004, China
*    Correspondence: jiangqingquan@xmut.edu.cn (Q.J.); yuhongxia@zjnu.cn (H.Y.)

**Abstract:** At present, there are few studies on breakthrough innovation (BI) involving entrepreneurial apprehension (EA). The purpose of this study is to identify how entrepreneurial apprehension affects breakthrough innovation. Based on the theory of planned behavior, the paper constructs a theoretical conceptual model of the influence of entrepreneurial apprehension and entrepreneurial strategic orientation (ESO) on breakthrough innovation and proposes corresponding research hypotheses, while using SPSS to conduct regression analysis of 216 valid questionnaires from high-tech enterprises in Yangtze River Delta. The results show that entrepreneurial apprehension can promote breakthrough innovation, but that entrepreneurial strategic orientation plays a partially mediating role in the process of entrepreneurial apprehension influencing breakthrough innovation. The incorporation of entrepreneurial apprehension into the study of breakthrough innovation complements and improves on the theory of entrepreneurial motivation for breakthrough innovation, and the study of entrepreneurial apprehension effectively expands the content of entrepreneurship theory. The paper concludes with the most important managerial implications and outlook for future research.

**Keywords:** entrepreneurship; entrepreneurial apprehension; entrepreneurial strategic orientation; breakthrough innovation

## 1. Introduction

In recent years, Chinese enterprises have been facing difficulties under the pressure of global technology competition, internal transformation, and upgrading, in attempting to reshape the technology track and market competition pattern through breakthrough technology to escape the predicament [1], and they are gradually becoming innovation leaders in some fields [2], but more Chinese enterprises are still buried in the industrial technology track that has been explored by foreign enterprises for incremental innovation [3–6]. Once foreign enterprises enter a new technology track through breakthrough technology change, they will be reduced to new followers again [7]. What exactly is the power to drive enterprises to make breakthrough innovations? Strategic orientation determines the development direction of enterprises, and the difference in the way in which enterprises carry out and implement technology innovation and obtain innovation benefits is determined by the differences in the choices of their strategic orientations [8]. Therefore, Chinese enterprises can fundamentally change from the role of "follower" to "leader" only through a different strategic orientation choice, i.e., moving from incremental innovation to breakthrough innovation [9]. However, the traditional strategic orientation determinism cannot fundamentally explain the innovation motives of different enterprises. The fundamental driving force needs to be found in the entrepreneurs who lead the strategic orientation of the enterprises. Entrepreneurial apprehension reflects the values and inner spiritual pursuits of the entrepreneurs, which can fundamentally explain the innovation

motives of the enterprises. By using a theoretical conceptual model/framework, this paper adopts a special perspective of entrepreneurial apprehension to study their relationship with breakthrough innovation.

As a distillation of the unique Chinese entrepreneurial trait, having a sense of worry and being able to think of danger in times of peace has received great attention from all walks of life in recent years. Although the sense of worry reflects the values and inner spiritual pursuits of entrepreneurs, it is different from the entrepreneurial spirit, in that its core characteristic is full of anxiety about the future development of the enterprise, especially about the future market changes and competitors. Therefore, it presents a strong sense of crisis, along with a strong sense of mission [10]. Many outstanding entrepreneurs have similar characteristics. As Huawei President Ren Zhengfei said in the famous "Huawei's winter" article: "Ten years I think every day is failure, blind to success, and there is no sense of honor, but a sense of crisis ...... failure this day is bound to come, we must be prepared to meet, this is I never waver See ......" [11]. A sense of worry is not unique to Ren Zhengfei; many of the world's top entrepreneurs have expressed similar views. Microsoft founder Bill Gates declared, "Microsoft is always only 18 months away from bankruptcy". Konosuke Matsushita, founder of Matsushita Electric, also said, "A persistent sense of crisis is the basis for making a company invincible". The list goes on and on. In general, however, most of the articles on entrepreneurs' sense of crisis are found in media reports and fragmentary expressions, and there is a lack of systematic research on it in academic circles, and so the study in this paper will help to expand upon this theme.

The factors influencing breakthrough innovation are numerous and complex, and scholars at home and abroad mainly focus on organization [12,13], technology [14,15], market [16,17], resources [18,19], and finance [20,21], among other dimensions. Studies have been made on the influencing factors of breakthrough innovation, and strategic orientation is one of the most important influencing factors. Strategic orientation refers to a strategic direction implemented by firms to guide their activities in order to achieve sustained better performance [22]; market orientation, technology orientation, and entrepreneurial orientation are the most commonly studied strategic orientations, and the role of these orientations on breakthrough innovation has been studied. Zhou et al. [23] found that market orientation and technology orientation have a greater effect on technology-driven breakthrough innovation, but no significant effect on market-driven breakthrough innovation. Shih [24] and Adams et al. [25] found that both market orientation and technology orientation have a positive effect on breakthrough innovation. Zhao et al. [26], Li et al. [27], and Mu et al. [28] found that entrepreneurial orientation is more inclined to breakthrough innovation, compared to market orientation and technology orientation. Priem et al. [29] found that among the many characteristics that influence managers' decisions, the most essential are the psychological factors and cognition of the manager. As the core decision makers, entrepreneurs' values and inner spiritual pursuits influence major decisions, including the choice of strategic orientation toward breakthrough innovations. In the process of choosing the strategic orientation of breakthrough innovation, entrepreneurs are both the decision makers of strategic orientation and the important organizers, promoters, and practitioners of breakthrough innovation, but breakthrough innovation is characterized by high investment, high risk, and high uncertainty [30], which requires entrepreneurs to have a sense of risky decision making, perseverance, and a sense of continuous self-transformation. As the research progresses, scholars are increasingly aware of the role of entrepreneurs' values and intrinsic spiritual pursuits on breakthrough innovation. Aragón-Sánchez et al. [31] classified entrepreneurs into three types based on different behavioral attitudes, namely defenders, explorers, and experiencers, arguing that breakthrough innovation patterns may vary slightly, depending on the entrepreneurs' attitudes. Feng et al. [32] and Liu et al. [33] classified entrepreneurs into career-seeking and wealth-seeking types, based on different pursuit goals, and found that a preference for wealth was negatively related to breakthrough innovation, and a preference for career was positively related to breakthrough innovation.

Although the above studies have classified the types of entrepreneurs based on different criteria and explored the influences of values and spiritual pursuits on breakthrough innovation, the role of entrepreneurial apprehension on breakthrough innovation has not been discussed. The purpose of this study is to fill the gap and identify their interactions. Based on the theoretical guidance of technological innovation, strategic management, entrepreneurship, and planning behavior, this study explores the role of entrepreneurial strategic orientation in the process of entrepreneurial apprehension affecting breakthrough innovation. Based on the perspective of entrepreneurial apprehension, this study can not only reveal the important variables affecting breakthrough innovation and their inner mechanisms of action, and further extend and improve the previous research results in this field, but it can also provide concrete guidance for the cultivation of anxious entrepreneurs and the formulation and implementation of breakthrough innovation strategies. The remainder of this research is organized in the following sections. The literature and hypotheses are reviewed, followed by the research methodology, and data analysis is reported. The study concludes with the discussion of study findings, theory contributions, management implications, limitations, and future research.

## 2. Literature Review

A review of literature shows that researchers have previously immersed themselves in attempting to understand entrepreneurial awareness-entrepreneurial strategic orientation-breakthrough innovation relationships respectively. Based on technological innovation, strategic management, entrepreneurship, and planning behavior, a variety of academic perspectives and frameworks have been explored in the field, such as market orientation [23–25,34,35], technology orientation [23–25], and entrepreneurial orientation [26–28,36,37]. Such approaches obscure our understanding of the evolution of the above factors as interactive networks. The following section adds more discussion for model development, and a new theoretical model and research hypotheses have been developed (Figure 1).

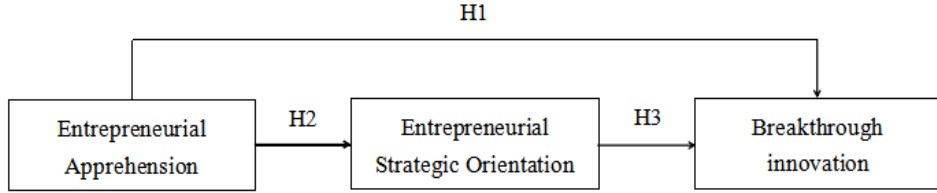

**Figure 1.** The conceptual model and research hypotheses on the relationship between entrepreneurial apprehension, entrepreneurial strategic orientation and breakthrough innovation.

*2.1. The Role of Entrepreneurial Apprehension on Breakthrough Innovation*

Furnham et al. [38] concluded that the values of managers determine the decisions and behaviors that they undertake in their organizations. Therefore, studying the original motivation of firms to carry out and to implement breakthrough innovations needs to be traced back to the values levels of decision makers. Having a sense of worry and being able to think in peace are important expressions of entrepreneurial values, which determine certain major decision-making behaviors of the firm. According to Ajzen's [39] theory of planned behavior (TPB), organizational behavior is influenced by the wills of organizational decision makers, which in turn are influenced by their perceptions and beliefs. The connotation of entrepreneurial apprehension shows that entrepreneurs are able to think of danger in times of peace, and the subjective norm of being apprehensive and being able to think of danger in times of peace will drive companies to continuously increase their R&D investment for breakthrough innovation [15]. For example, when Jack Ma participated in the "Economic Situation Analysis and Research Conference of Zhejiang Entrepreneurs", he emphasized that "leaders at all levels of the enterprise should have vision and broad-mindedness, form their own perception of the external environment, always have a sense of worry, and deeply understand that living has become the norm that

enterprises must face ...... sunny days to repair the roof is the most important. When the business is profitable and morale is high, it is instead the time when the business must be reformed and innovated" [40]. At the same time, entrepreneurs with a sense of worry have a stronger sense of responsibility for their businesses. When they achieve initial success in business, they will not have the mentality of "taking the good with the bad" or "getting rich and getting comfortable", but they will instead have the self-driven mentality of "the more successful you are, the greater the responsibility" [41]. This endogenous motivation will inspire them to continuously expand and to upgrade their social networks to find breakthrough innovation opportunities, so as to better reward their employees, customers, and society at large [42,43].

Furthermore, entrepreneurs with a sense of concern focus on a wider range of interest groups than ordinary entrepreneurs, and the wider their social networks [44], the stronger their ability to identify opportunities from the external environment [45], and the more resources they have to devote toward breakthrough innovation activities [46,47]. For example, Huawei surpassed Ericsson in sales revenue in 2013 to become the world's largest telecom equipment supplier and a leader in key technologies. When public opinion poured the praise of "the world's first" to Huawei, "Will Huawei be the next to fall" once again triggered Huawei cadres at all levels to think about the potential crisis in the future: constantly seeking partners in the fields of technology development, production, and sales and services and increasing R&D investment [48]. Since 2013, Huawei has invested more in R&D than net profit for seven consecutive years, and it has ranked first in the world in terms of the number of patent applications at the European Patent Office (EPO) in 2019, leading the major technology giants, with 3524. Through the above analysis, the following hypothesis is proposed.

**H1.** *Entrepreneurial apprehension contributes to breakthrough innovation.*

*2.2. The Role of Entrepreneurial Apprehension in the Strategic Orientations of Entrepreneurs*

As the core decision makers of a company, entrepreneurs influence the major decisions of the company with their values and intrinsic spiritual pursuits, which include the development of strategic orientation [41]. From a temporal perspective, entrepreneurial strategic orientation does not involve future decisions, but rather the futurity of current decisions [49]. What entrepreneurs should do is use the information that they have to make sound decisions in the present, and to prepare them for the future [50]. What motivates entrepreneurs to be well prepared for the future? In the end, it is a sense of worry that is rooted in their hearts [51]. Successful decision makers want to make their business grow in the long run, but they are afraid that sudden changes will be detrimental to their business, and that they will make mistakes or omissions [52]. Hence, they look to the future, prepare early, and carefully monitor the implementation of their strategies.

Entrepreneurial strategy-oriented implementation and control are two different concepts [53]. Both are components of strategic management, and they co-exist in the day-to-day management of a business. In the process of business operation, entrepreneurs have to consider whether many factors, such as technology development, production, marketing, finance, human resources, etc., are closely coordinated, to ensure the smooth implementation of the strategy; whether there are deviations in these factors, whether the strategy implementation is in line with the strategic objectives [54], and whether the current strategy is in line with the current market environment changes, etc. [55]. Therefore, the strategy-oriented implementation and control approach also involves this sense of worry. The cultivation of this sense of worry comes from an excellent corporate culture, which is built by entrepreneurs [56]. In order to establish a perfect crisis management team, the entrepreneur must be the one to outline the organization [57]. It has been proven that a flawless business strategy can end up being worthless if it is poorly implemented and monitored. For the daily operation of the company, the entrepreneur must have a clear head and be cautious at all times [58]. Only in this way can the operator control the internal

factors of the company so that the company's decisions will be less faulty and erroneous, while ensuring that the company will better adapt to changes in the external environment, and that its operations will develop in the right direction. Through the above analysis, the following hypothesis is proposed.

**H2.** *Entrepreneurial apprehension contributes to entrepreneurial strategic orientation.*

### 2.3. The Role of Entrepreneurial Strategic Orientation on Breakthrough Innovation

Progressive technology innovation is an adjustment and an improvement on the existing foundation, which is a small deviation from existing organizational practices and an optimization of organizational practices. Breakthrough innovation, on the other hand, produces fundamental changes in organizational behavior and is a large deviation from the existing organizational practices [59]. The difference between these two types of innovation is that breakthrough innovation contains more new knowledge and technology than progressive technology innovation, and that mutation innovation often leads to the emergence of new products and processes, while progressive technology innovation mostly shows the improvement and optimization of products and processes on the basis of the original ones. On the other hand, breakthrough innovation is riskier than incremental technology innovation. Generally speaking, breakthrough innovation requires a higher amount of investment capital, technology, and management than incremental technology innovation, but the success rate is lower than incremental technology innovation [60]. In a dynamically changing environment, the dilemma facing managers and organizations is clear: In the short term, there must be a constant focus on incremental technology innovation, and on increasing the strategic structure and cultural fit. However, this is not sufficient for sustained success, and for the long term, the inertia that makes organizations successful must be broken, and breakthrough innovation must be implemented [61–63].

Differences in the way in which firms carry out the implementation of technology innovation and reap the benefits of innovation are caused by differences in the choices of their strategic orientations [8]. Under different strategic orientations, firms differ in the allocation patterns of their own resources, which can affect the way in which their technology innovation behaviors are accomplished, and this can make their competitive advantages appear differentiated [64]. Entrepreneurial strategic orientation is a culture that pursues advancement, pioneering, and innovation, as well as an atmosphere that promotes learning and encourages innovation, which makes the whole company committed to innovation and creativity, and accomplishes the strategic goal of remediation through continuous search and exploration [65]. Entrepreneurial orientation emphasizes the creation of change and making differences [66]. Under entrepreneurial strategic orientation, firms tend to break the existing operational practices, as well as the normative guidelines of cooperation, to fully stimulate organizational creativity and gain competitive advantage in a new way [67]. Through the above analysis, the following hypothesis is proposed.

**H3.** *Entrepreneurial strategic orientation contributes to breakthrough innovation.*

## 3. Research Methodology

### 3.1. Sample and Data Background

In this paper, entrepreneurs are defined as the core founders of a business (involved in starting the business and having the greatest control), while still being the actual head of the business at present. The survey was conducted in the form of written questionnaires and in-person interviews with high-tech enterprises in the Yangtze River Delta. A total of 459 questionnaires were distributed, and 258 questionnaires were collected. After excluding the questionnaires in which the founder was not the actual person in charge of the company, or the questionnaires that did not meet the requirements, and those that were not filled out well, 216 valid questionnaires were obtained, accounting for 47.06% of the total.

(1) Enterprise level. From the viewpoint of enterprise establishment, the majority of enterprises were 2–5 years old (47.69%), indicating that most of the sample enterprises were start-ups and more mature enterprises. From the viewpoint of enterprise size, the highest percentage of enterprises comprised 300 or less employees (62.50%); from the viewpoint of annual sales (AS), the percentage of sales below 10–100 million was 50.60%. From the viewpoint of industry characteristics, Sino–foreign joint ventures (SFJV) and local enterprises (LE) accounted for 82.41%. From the perspective of the industry affiliation, electronic information (EI), new energy and energy-saving (NEES), resources and environment (RE), and high technology service (HTS) enterprises accounted for a relatively large proportion of a total of 55.63%, while biology and new medicine (BNM), aerospace (AS), new materials (NM), advanced manufacturing automation (AMA), and other types of enterprises were more evenly distributed. The characteristics of the questionnaires are shown in Figures 2–6.

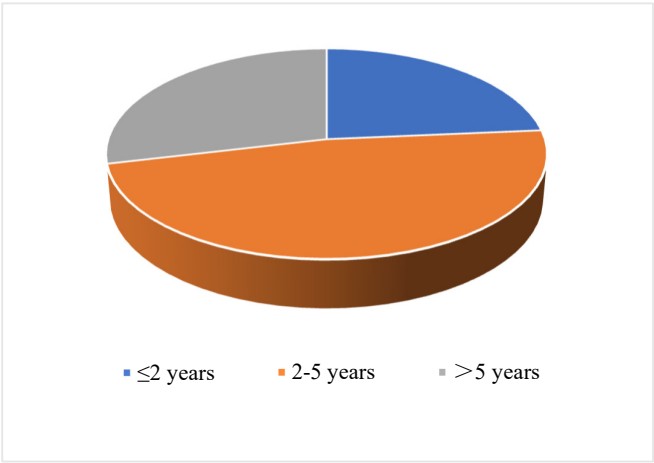

**Figure 2.** Enterprise establishment characteristic.

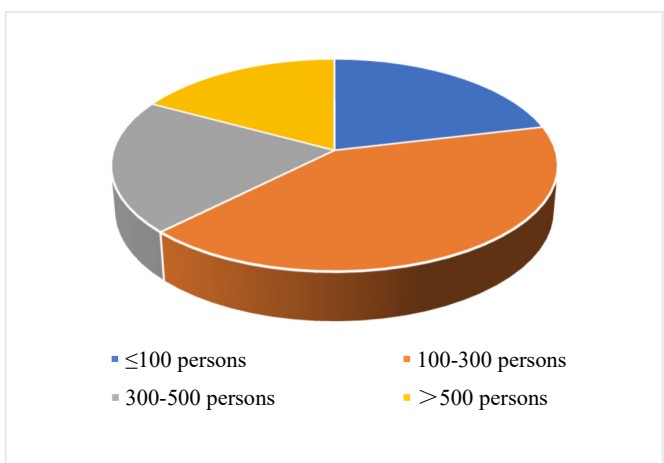

**Figure 3.** Enterprise size characteristic.

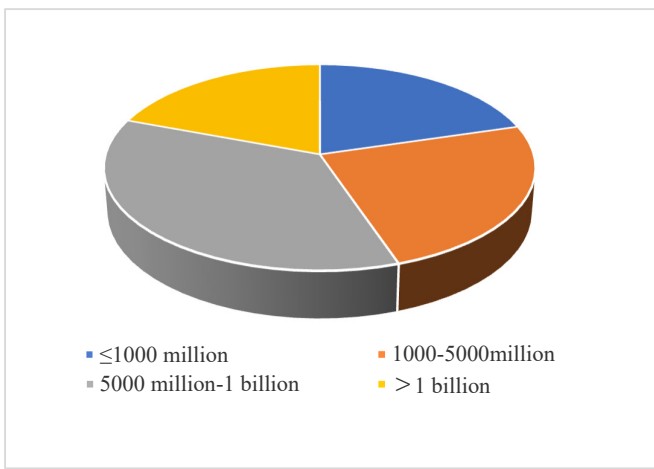

**Figure 4.** Annual sales characteristic.

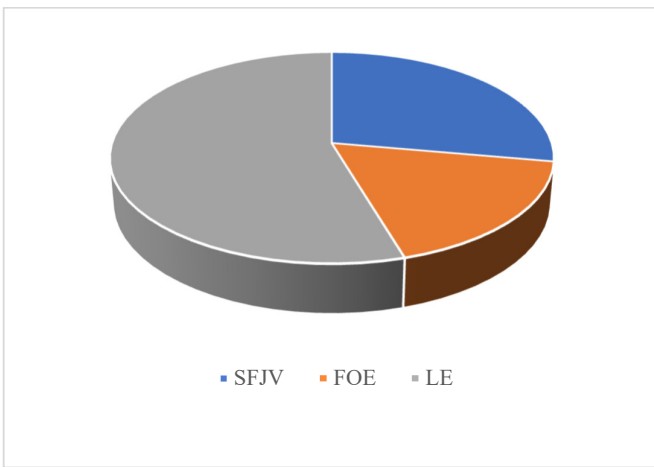

**Figure 5.** Industry characteristic.

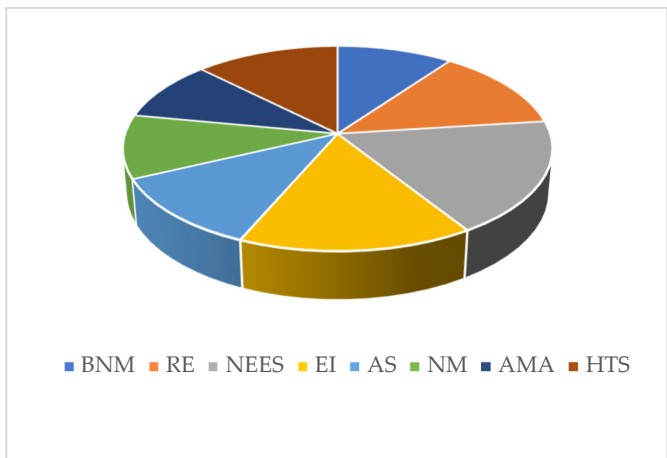

**Figure 6.** Industry affiliation characteristic.

(2) Entrepreneurial level. In terms of the gender of the sample entrepreneurs, men accounted for 91.03%; in terms of age, the four age groups above 30 years old were more evenly distributed, and those below 30 years old accounted for the minority, at 5.12%; in terms of education level, bachelor's and master's degrees accounted for more, at 33.25% and 31.39%, respectively, and those at college and below accounted for 5.38% and 2.68%, respectively, which is shown in Figure 7.

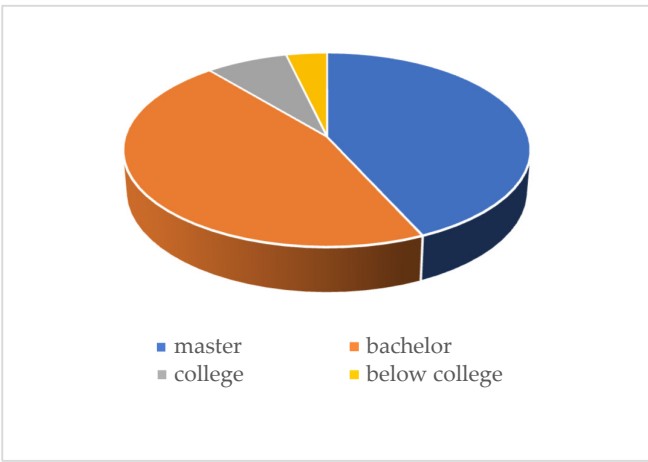

**Figure 7.** Education level characteristic.

*3.2. Variable Measures*

Based on Teece's [68,69] dynamic competence theory, it is proposed to measure entrepreneurial apprehension with a total of 10 questions in three dimensions: identifying environmental changes, communicating apprehension, and formulating countermeasures. Regarding entrepreneurial strategic orientation, first, we referred to Lumpkin and Dess [65,70] to define the connotation of entrepreneurial orientation, and to measure entrepreneurial strategic orientation based on the three dimensions of strategy formulation, implementation, and control, with a total of 12 items. Regarding the breakthrough innovation of enterprises, we combined the work of Jiang et al. [1], Atuahene-Gima [71], Keupp and Gassmann [72], Forés and Camisón [73], and Gao and Liu [74]. A total of four items from two dimensions of innovation quantity and innovation capability were used to measure corporate breakthrough innovation.

In order to control the influences of other factors on the dependent variable, this paper synthesizes the existing literature and includes entrepreneur gender, entrepreneur education, firm age, firm size, and industry characteristics as control variables. Except for these five control variables, all of the above questionnaires were measured in the form of a five-point Likert scale, in which subjects were asked to give their level of agreement with each item according to their own organizations, with 1 indicating "strongly disagree" and 5 indicating "strongly agree".

## 4. Data Analysis

*4.1. Reliability Validity Analysis*

In this paper, Cronbach's alpha was used to test the reliability of the questionnaire, and the results showed that the Cronbach's alpha values of entrepreneurial apprehension, entrepreneurial strategic orientation, and corporate breakthrough innovation were 0.931, 0.859, and 0.836, respectively, which were all greater than 0.8, indicating a good internal consistency of the variables.

In order to ensure the high validity of the questionnaire, the research team conducted a pre-study and made several adjustments and modifications through expert consultation before the questionnaire was distributed, and they also used exploratory factor analysis to classify the structure of the main variables. The results show that the KMO values of entrepreneurial apprehension, entrepreneurial strategic orientation, and corporate breakthrough innovation were respectively 0.801, 0.846, and 0.857, and the significance value of Bartlett's sphericity test was 0.000, which indicated a high significance and the suitability of exploratory factor analysis.

### 4.2. Correlation Analysis of Variables

Correlation analysis is the degree of correlation between variables and does not involve causality, which is the basis and prerequisite for regression analysis, and can initially determine the reasonableness of the model and hypothesis setting [75]. Table 1 shows the results of correlation analysis between variables, and the critical values of correlations between the variables are all less than 0.4 and reach the significance level, which indicates that the questionnaire has good discriminant validity, and this provides a good basis for the further exploration of interrelationships.

**Table 1.** Correlation coefficients between variables.

| Variables | Average Value | Standard Deviation | EA | ESO | BTI |
|---|---|---|---|---|---|
| EA | 4.025 | 0.763 | 1 | | |
| ESO | 2.656 | 0.624 | 0.256 ** | 1 | |
| BI | 3.753 | 0.954 | 0.308 *** | 0.169 ** | 1 |

Note: ** $p < 0.01$, *** $p < 0.001$.

### 4.3. Regression Analysis

(1) Hypothesis validation analysis. The correlation analysis shows that the correlation between the above variables is initially consistent with the conceptual model, which indicates that the hypothesis and the model proposed are more reasonable, but the correlation between the variables cannot explain the causal relationship between the variables [41], so this paper continued the regression analysis of each variable to explain the causal relationship between the variables and the magnitude of the influence. According to the suggestions of Chang et al. [76], this paper tried to fit with linear regression, logit, quadratic curve regression, and cubic curve regression models, and found that all of them passed the significance test. As shown in Table 2, these four regression models are statistically significant. The sizes of the decidable coefficients of the quadratic, cubic, and linear models did not change much, and the result shows that the linear model has the largest F-value and fits better compared to the other curves. This means that Hypothesis H1 was verified, i.e., the stronger the entrepreneur's sense of worry, the higher the level of the breakthrough innovation of the enterprise.

**Table 2.** Model summary and parameter estimates.

| Equation | Dependent Variable: Breakthrough Innovation | | | | | | | | |
|---|---|---|---|---|---|---|---|---|---|
| | Model Summary | | | | | Parameter Estimates | | | |
| | $R^2$ | F | df1 | df2 | Sig | Constants | b1 | b2 | b3 |
| Linear | 0.892 | 1257.235 | 1 | 214 | 0.000 | −0.658 | 2.100 | | |
| Logarithmic | 0.875 | 1109.458 | 1 | 214 | 0.000 | −1.265 | 4.350 | | |
| Two times | 0.881 | 783.795 | 2 | 213 | 0.000 | −1.286 | 1.623 | 0.087 | |
| Three times | 0.868 | 697.624 | 3 | 212 | 0.000 | −0.563 | 0.807 | −0.621 | −0.035 |

Note: The independent variable is entrepreneurial apprehension.

(2) Mediating effect analysis. We adopted the causal stepwise regression test proposed by Baron and Kenny [77] to test the mediating effect. The test is divided into three steps: The first step is to analyze the effect of the independent variable on the dependent variable, and to test the significance of the regression coefficient. The second step is to analyze the effect of the independent variable on the mediating variable and to test the significance of the regression coefficient. The third step is to analyze the regression of the independent variable on the dependent variable after adding the mediating variable, and to test the significance of the regression coefficient. When the effect of the independent variable on the dependent variable disappears, it means that the mediating variable plays a fully mediating role, and if the effect is still significant, it is partially mediating.

From the previous analysis, it can be found that the first two steps in the causal stepwise regression test method have been verified: the influence of entrepreneurial preference awareness on enterprise breakthrough innovation is significant, i.e., the independent variable influences the dependent variable; the influence of entrepreneurial preference awareness on entrepreneurial strategic orientation is significant, and the influence of entrepreneurial strategic orientation on enterprise breakthrough innovation is also significant, i.e., the independent variable influences the mediating variable, and the mediating variable influences the dependent variable.

According to the regression results of Model 2 in Table 3, it can be found that the $p$-value is less than 0.05, and thus Hypothesis H2 is verified; then, according to the regression results of Model 3 in Table 3, it can be found that when entrepreneurial strategic orientation is added as a mediating variable in the regression model of the influence of entrepreneurial preference awareness on the breakthrough innovation of enterprises, the coefficient of the influence of entrepreneurial preference awareness on the breakthrough innovation of enterprises is reduced from 1.356 to 1.028, but the significance still exists, indicating that the third step is completed. The strategic orientation of entrepreneurs plays a partially mediating role in the relationship between entrepreneurial preference awareness and entrepreneurial breakthrough innovation, and thus Hypothesis H3 is verified.

**Table 3.** Regression results of EA and ESO on BTI.

| Variables | Mediating Variable: ESO | | |
| --- | --- | --- | --- |
| | Model 1 BI | Model 2 ESO | Model 3 BI |
| Entrepreneur Gender | 0.103 | 0.023 * | 0.045 |
| Entrepreneur Education | −0.018 | −0.200 * | 0.006 |
| Entrepreneur Age | 0.017 | 0.051 | 0.023 |
| Year of Establishment | −0.002 | 0.024 * | 0.009 |
| Enterprise Size | 0.017 | 0.025 ** | 0.011 |
| EA | 1.356 *** | 0.758 *** | 1.028 *** |
| ESO | | | 0.362 *** |
| R2 | 0.752 | 0.647 | 0.815 |
| Adjust R2 | 0.763 | 0.625 | 0.762 |
| F | 134.586 | 25.268 | 117.265 |
| Sig. | 0.000 | 0.000 | 0.000 |

Note: * $p < 0.05$, ** $p < 0.01$, *** $p < 0.001$.

## 5. Discussion and Conclusions

### 5.1. Discussion of Study Findings

(1) The study findings show that entrepreneurial apprehension can indeed promote breakthrough innovation. Although scholars at home and abroad have recognized the important role of the entrepreneurial values and spiritual pursuits on breakthrough innovation and have classified the types of entrepreneurs based on different criteria, they have neglected the role of entrepreneurial apprehension on breakthrough innovation. In fact, entrepreneurs who have a sense of worry and can think of danger in times of peace have a stronger sense of responsibility for their enterprises. When they achieve initial success in business, they do not have the mentality of "taking the good with the bad". On the one hand, this self-drive will drive the enterprise to continuously increase R&D investment for breakthrough innovation, and on the other hand, it will continuously expand and upgrade its social network to look for breakthrough innovation opportunities, so as to better the enterprise. (2) At the same time, the study findings show that entrepreneurial strategic orientation plays a partially mediating role in the process of entrepreneurial apprehension influencing breakthrough innovation. It is the high-level spiritual pursuit embedded in the sense of anxiety that motivates entrepreneurs to strongly want to make the enterprise develop in the long run, and even to have a strong strategic orientation in influencing the direction of enterprise development, so that it promotes breakthrough innovation. Otherwise,

they will still become caught up in the industrial technology track that has been explored by foreign enterprises for incremental innovation, and once foreign enterprises enter a new technology track through the breakthrough technological change, they will be reduced to a new follower again. Therefore, the research on the influencing factors of enterprise breakthrough innovation should not only stay at the level of organization, technology, market, resources, finance, etc., proposed by traditional research perspectives. These external influencing factors cannot really explain the deeper reasons for enterprise breakthrough innovation. The personal spiritual pursuits and behavioral goals of entrepreneurs are the real internal fundamental driving forces of enterprise breakthrough innovation, and entrepreneurial apprehension can fundamentally explain enterprises' motivation to innovate.

*5.2. Theory Contributions and Management Implications*

The contributions of this study are: (1) This paper incorporates entrepreneurial apprehension into the framework of breakthrough innovation research, which complements and improves on the theory of entrepreneurial motivation for breakthrough innovation. Previous studies have often summarized the motivation of breakthrough innovation as being technology-driven, demand-pulled, or a combination of both, but these factors only remain on the surface, while a sense of worry is the embodiment of entrepreneurs' internal values and spiritual pursuits, and the breakthrough innovation driven by such internal values and spiritual pursuits is a type of conscious and spontaneous behavior, which brings long-term and sustainable impact to the enterprise. The long-term and sustainability of its impact on the enterprise far exceeds the external driving force. (2) The study of entrepreneurial apprehension has effectively expanded the theory of entrepreneurship. Entrepreneurship is a collection of special skills of entrepreneurs. Most of the existing studies consider entrepreneurship and innovation as the core of entrepreneurship, while risk-taking ability, market opportunity recognition, and risk-taking spirit are important characteristics of entrepreneurship. It can be said that the current definition of entrepreneurship remains at the surface level, while entrepreneurial apprehension reaches a deeper level of cognition, which is a further sublimation of the study of the connotation of entrepreneurship.

Management implications from this study: (1) For entrepreneurs, it is important to continuously improve their spiritual cultivation and competence cultivation, and to strive to enhance their ability to identify environmental changes, convey a sense of worry, and formulate countermeasures. With the increasing changes in the technology and market environment in China and abroad, enterprises have to continuously improve their ability to adapt to environmental changes, and as the core person of the enterprise, entrepreneurs must also have the spirit, awareness, literacy, and ability that this role should have, as it is related to the success or failure, and the long-term development of the enterprise. The more complex the business environment is, the higher the requirements for the entrepreneur's spiritual level, ability to identify environmental changes, convey the sense of worry, and formulate countermeasures. (2) Strategic orientation determines the direction of enterprise development. Compared with technology orientation and market orientation, entrepreneurial orientation is more focused on the long-term developments of enterprises, and it tends to involve the implementation of breakthrough innovations. In this process, entrepreneurs are not only the makers, but also the implementers and controllers of breakthrough innovation strategies, which puts high demands on the abilities of entrepreneurs to be strategically oriented.

*5.3. Limitations and Future Research*

Shortcomings of this study: (1) This study focuses on three structural dimensions of entrepreneurial apprehension, namely identifying environmental changes, conveying apprehension, and formulating countermeasures. However, in fact, entrepreneurial apprehension also includes other components, which may have more or less influence on breakthrough innovation. Although this measurement method is generally recognized and accepted in behavioral science, it is not the actual data, and it still has a certain distance

from reality, such that it can be combined with financial data in the future to make the data more effective and objective. (3) This study only preliminarily explored the role of entrepreneurial strategic orientation in the process of entrepreneurial apprehension influencing breakthrough innovation, but future research can explore the mechanism of entrepreneurial apprehension influencing breakthrough innovation based on entrepreneurial risk-taking, stress response, employee creativity, organizational identity, etc. Meanwhile, although the number and achievements of entrepreneurs in Zhejiang are ahead of the whole country, and entrepreneurial apprehension has better performance than enterprises in other regions, it still does not represent the whole situation, and future research should appropriately expand the scope of the test in order to increase the generalizability of the research findings.

**Author Contributions:** Conceptualization, Q.W.; investigation, Q.J.; data curation, H.Y. All authors have read and agreed to the published version of the manuscript.

**Funding:** This article was sponsored by the Project supported by National Social Science Foundation of China (Grant No. 20BGL132), Humanities and Social Sciences Research Project of the Ministry of Education (Grant no. 20YJC630199), Soft Project of Zhejiang Province (Grant No. 2021C35089), Quzhou University Research Start-up Fund (Grant No. BSYJ202201).

**Institutional Review Board Statement:** Not applicable.

**Informed Consent Statement:** Not applicable.

**Data Availability Statement:** Not applicable.

**Acknowledgments:** The authors appreciate the anonymous reviewers for their constructive comments and suggestions that significantly improved the quality of this manuscript.

**Conflicts of Interest:** The authors declare no conflict of interest.

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
