# Peer review of "Analysis of the Influence of Entrepreneurial Apprehension and Entrepreneurial Strategic Orientation on Breakthrough Innovation"

_sustainability, doi:10.3390/su15097320_

Round 1

Reviewer 1 Report

Thank you very much for allowing me to review this paper. This an interesting paper where you have proposed a theoretical conceptual model of the influence of entrepreneurial apprehension and entrepreneurial strategic orientation (ESO) on breakthrough innovation, and proposes corresponding research hypotheses, while conducting an empirical analysis using 216 valid questionnaires from high-tech enterprises in Yangtze River Delta. The purpose of this study could be written more clearly in the abstract and introduction sections of the paper. Certainly, the authors have made significant discussions about its impact and made a significant contribution to relevant academic and practitioner literature.  However, I think your introduction should have a few more citations to justify the problem statement properly. Although the concept of this paper is good, it has not followed the submission guidance for Sustainability journal by MDPI. Particularly, the structure of this paper where you have introduction and then a Research Overview section 2. Please follow some papers from the Sustainability journal by MDPI. The methodology section of this paper is slightly descriptive in nature and a better justification for chosen methodology is required. Results are OK but a critical discussion is expected at this level. There are some good recommendations and future research directions with some limitation/shortcomings at the end.

Reviewer 2 Report

The manuscript “Entrepreneurial Apprehension, Entrepreneurial Strategic Orientation, and Breakthrough Innovation” approaches a topic of interest to this Journal. The document presents a well-organized and comprehensive study on entrepreneurship. The research design and hypotheses are clearly stated. However, the H1 description is missing in the text. The results are presented in a concise manner, but the discussion is absent. I recommend building a text to connect the results with the theoretical findings.

Reviewer 3 Report

Dear Authors,

Please find below and attached my comments and suggestions for your work.

Good luck!

Kind regards,

The Reviewer

Review Report Form

Partea superioară a machetei

Journal: Sustainability (ISSN 2071-1050)

Manuscript ID: sustainability-2304494

Type: Article

Title: Entrepreneurial Apprehension, Entrepreneurial Strategic Orientation, and Breakthrough Innovation

Authors: Qiliang Wang , Qingquan Jiang * , Hongxia Yu *

Topic: Decision Making Behaviors in Management and Marketing

Submission Date: 10 March 2023

Dear Authors,

I have carefully analyzed your article entitled “Entrepreneurial Apprehension, Entrepreneurial Strategic Orientation, and Breakthrough Innovation”.

Congratulations for your work and valuable insights reflected in the content of the manuscript!

The structure of my Review Report Form takes into consideration two sections, namely: (A.) General overview of the article and strong points; and (B) Suggestions meant to improve your current manuscript.

(A.) General overview of the article and strong points:

Ø  General background of the study & Aim of the study: The authors have mentioned that, at present, there are few studies on breakthrough innovation (BI) involving entrepreneurial apprehension (EA) and its strategic orientation factors.

Ø  Research methodology used: The authors have pointed out that based on the theory of planned behavior, the paper constructs a theoretical conceptual model of the influence of entrepreneurial apprehension and entrepreneurial strategic orientation (ESO) on breakthrough innovation, and proposes corresponding research hypotheses, while conducting an empirical analysis using 216 valid questionnaires from high-tech enterprises in Yangtze River Delta.

Ø  Results of the study: In terms of the results of this current study, it ought to be mentioned that that the authors found that entrepreneurial apprehension can promote breakthrough innovation, but that entrepreneurial strategic orientation plays a partially mediating role in the process of entrepreneurial apprehension influencing breakthrough innovation. Moreover, the authors have pointed out that the incorporation of entrepreneurial apprehension into the study of breakthrough innovation complements and improve on the theory of entrepreneurial motivation for breakthrough innovation, and the study of entrepreneurial apprehension effectively expands the content of entrepreneurship theory. Furthermore, the paper concludes with managerial implications for practice, and an outlook for future research.

(B) Suggestions meant to improve your current manuscript:

Distinguished Authors I would kindly like to suggest the following aspects:

(1.) Closely analyzing the article, since there are some English language improvements and slight corrections that need to be taken care of. Thus, my recommendation would be to carefully proofread the entire manuscript. 

(2.) Also, I have closely analyzed the format of the article, in order to check whether it follows the guidelines which are specific to the publisher. Thus, I have noticed that the current form of your work needs improvement in this regard. So, my kind suggestion is to closely analyze again the guidelines belonging to the publisher, since the article should fit exactly the publisher’s guidelines. For instance, the keywords, the subsections, the references, currently do not fit the style and the requirements of the publisher. Also, it would be highly recommendable to include in the abstract of your study more highly relevant details that refer to the research objectives and the methodology used. This would definitely be considered a plus for your scientific work.   

(3.) I would highly recommend offering additional explanations for the reasoning of the methodology used, in order to support the validly of the data as well as the importance of the results obtained at all levels.  

(4.) In continuation, the suggestion would also be inserting in your article a few ideas concerning the correlation between effects of the COVID-19 pandemic and the COVID-19 global crisis, sustainability and sustainability assessment, Sustainable Development Goals, while focusing on entrepreneurial apprehension, entrepreneurial strategic orientation, and breakthrough innovation, since these are key focuses these days. In this context, I had the chance to read a few interesting scientific works recently, among which I would like to mention: Popescu, C. R. (2021). Impact of Innovative Capital on the Global Performance of the European Union: Implications on Sustainability Assessment. In C. Popescu (Ed.), Handbook of Research on Novel Practices and Current Successes in Achieving the Sustainable Development Goals (pp. 90-124). IGI Global. https://doi.org/10.4018/978-1-7998-8426-2.ch005; OECD. Measuring the Impacts of Business on Well-Being and Sustainability. https://www.oecd.org/statistics/Measuring-impacts-of-business-on-well-being.pdf; OECD. 2022. Toward sustainable economic development through promoting and enabling responsible business conduct. https://www.oecd-ilibrary.org/sites/f7813858-en/index.html?itemId=/content/component/f7813858-en.

Dear Authors, congratulations once again for your work and valuable insights reflected in the content of the manuscript, and I hope my comments will be of value to you!

Kind regards,

The Reviewer

Partea inferioară a machetei

Reviewer 4 Report

Title could perhaps be changed to reflect the findings but this is just a suggestion

My understanding of the paper is that entrepreneurs sense of worry impacts the level of breakthrough innovation of their enterprise and their entrepreneurial apprehension affects their entrepreneurial strategic orientation which partially mediates breakthrough innovation

Background knowledge in the introduction is a little weak.

Methodology appears sound

Abstract

Missing in the introduction is definitions/intro/explanation of Entrepreneurial apprehension, Entrepreneurial strategic orientation and breakthrough innovation. “catcher versus leader” needs to be defined for someone who is new to the theory of entrepreneurship topic. (insert Lines 72 to 75 here as it defines/introduces Strategic orientation, same for Lines 204 to 216 placing technology innovation and breakthrough innovation within an context and provides background information.)

Line 20: instead of “The paper concludes with managerial implications for practice, and an outlook for future research.” Insert the most important managerial implications and outlook for future research.

Introduction

Lines 27 to 28 – what evidence have you inserted proving these companies have been experiencing difficulties. At the moment it is an opinion.

Line 32 makes no sense: “They have not yet gotten rid of the strange circle 32 of introducing backward and then introducing backward again”.

Lines 42 to 46 sentence is too long. Break it up into shorter sentences as I am getting lost reading this.

Lines 47 to 48, add in ‘. using a theoretical conceptual model/framework.’

Lines 56 to 62, insert in-text references of the entrepreneurs speeches.

Research overview

Lines 84 to 86 , Line 105 to 110 – requires intext reference evidence.

Line 115 ..’ this paper explores using a conceptual model/framework…

Conceptual Model/Research Hypothesis

Lines 137 to 142  and lines 154 to 167  and Line 174 to 180 Line 193 to 200 - needs an intext reference to support these statements as fact and not hearsay - also sentences are too long and needs to be broken into smaller sentences for ease of the reader to absorb the information

Research Methodology

Lines 252 to 269- some of the data would be better presented in Charts rather than written explanation

Main Research Findings

Perhaps you can redraw the model and show the partial mediating role visually to reflect the findings.

Only suggstion when it comes to the written work is shorter sentences by breaking up some of the very long sentences so it is easier for the reader to digest, some diagrams or charts

Conclusions appear justified

Reviewer 5 Report

The paper is quite ineteresting, with academic merit.

Some suggestion that could improve the work.

- line 105.: In general, there is a large body of literature on entrepreneurial awareness, entrepreneurial strategic orientation, and breakthrough technological innovation,...

No referrences exist. Authors should provide some representative researches they are referred to. They extensively referred to researches on other issues on innovation, however no such references exist here for awareness and innovation.

- The authors use specific examples of real life, maingly through the words of managers etc. However, a more scientific form is also needed, with researches in that field, to increase the overall academic merit of the paper. In general, the text in some parts (especially Section 3) needs to become more"scientific", to be enriched with referrences in each field described in theoretical background.

- Line 201 (and 167-168). The authors begin from Hypothesis 2. Hypothesis 1 is missing!!

- Fig.1 should be referred (even with a sort sentence) in the text..

- The authors should provide some details on why they choose to use to different regression models. For instance, logistic regressions are more appropriate for binary dependent variables. Why for the breakthrough innovation they use both regression models?

- The software used for the econometric analysis should be mentioned (SPSS probably?)

- The last part (results and outlook) is divided in many autonomous parts. Instead of that, it would be better i believe, to be separated in a discussion section and then conclusions. In the former, some connections with the theoretical background could be also included.

Round 2

Reviewer 1 Report

Thank you very much for the opportunity to review this above-titled paper again. There are many things I like about the paper. First, the paper touches on an important subject in the literature which has been overlooked by researchers. I am happy with the changes. 

Author Response

Thank you again for your valuable suggestions, which are very useful for the improvement of paper.

Reviewer 2 Report

Dear Authors,

The changes embedded in second submission meet the review comments.

I wish high impact to the publication.

Author Response

(The authors gave the same response as above.)

Reviewer 4 Report

Line 212 - Change the term "more risky" to "riskier"

Line 256 - Amend the term "millions" to "million"

Line 266 - Amend thet erm in the diagram "millions" to "million"

Line 328 - remove the term "vary"

Reviewer 5 Report

its ok now

Author Response

(The authors gave the same response as above.)
